# Magnetic Data Correction for Fluxgate Magnetometers on a Paramagnetic Unmanned Surface Vehicle: A Comparative Analysis in Marine Surveys

**DOI:** 10.3390/s25144511

**Published:** 2025-07-21

**Authors:** Seonggyu Choi, Mijeong Kim, Yosup Park, Gidon Moon, Hanjin Choe

**Affiliations:** 1Department of Geological Sciences, Pusan National University, Busan 46241, Republic of Korea; 2Korea Institute of Ocean Science and Technology, Pohang 37553, Republic of Korea; 3Oceantech Co., Ltd., Goyang 10440, Republic of Korea

**Keywords:** marine magnetic survey, fluxgate magnetometer, viscous magnetization, magnetic correction, paramagnetic, Unmanned Surface Vehicle

## Abstract

**Highlights:**

**What are the main findings?**
Paramagnetic USV offers a cost-effective solution for marine magnetic surveys by significantly reducing operational expenses compared to conventional research vessels.Paramagnetic hull effectively reduces magnetic noise and shortens the duration of viscous magnetization.

**What is the implication of the main finding?**
Comparative analysis of the correction methods from various platforms highlights that employing a 9-component susceptibility tensor is the most robust.Validated relative accuracy but revealed consistent ~200 nT offset vs. towed scalar data.

**Abstract:**

Unmanned Surface Vehicle (USV) offers a cost-effective platform for high-resolution marine magnetic surveys using shipborne fluxgate magnetometers. However, platform-induced magnetic interference and electromagnetic interference (EMI) can degrade data quality, even with paramagnetic hulls. This study evaluates fluxgate magnetometer data acquired from a paramagnetic-hulled USV. Noise characterization identified EMI and maneuver-induced high-frequency noise, the latter of which was effectively reduced through low-pass filtering. We compared four different correction approaches addressing both vessel attitude and magnetization. The results demonstrate that the paramagnetic hull significantly reduces magnetic interference and shortens the duration of viscous magnetization (VM) effects caused by eddy currents in the platform, compared to conventional ferromagnetic vessels. Nonetheless, residual magnetization from onboard ferromagnetic components still requires correction. A method utilizing all nine components of the susceptibility tensor demonstrated improved accuracy and stability. Despite corrections, low-frequency VM-related noise during azimuth changes and a consistent absolute offset (~200 nT) remain when compared to towed scalar magnetometer data. These findings validate the use of paramagnetic USV for vector magnetic surveys, highlighting their benefit in VM mitigation while emphasizing the need for further development in VM correction and offset correction to achieve high-precision measurements.

## 1. Introduction

Marine magnetic surveys, a non-destructive and cost-effective method, play a crucial role in a wide range of geophysical applications [1,2,3,4]. These surveys are conventionally performed using scalar or fluxgate magnetometers installed on research vessels. Scalar magnetometers provide high-precision measurements of the total magnetic field intensity, while fluxgate magnetometers are valued for their ability to acquire vector components of the magnetic field. These vector magnetic anomalies can clarify the various interpretations arising from scalar magnetic anomaly analysis [5,6]. However, compared to towed scalar magnetometers, fluxgate magnetometers mounted on ships exhibit lower resolution. This is due to electromagnetic interference originating from the ship’s ferromagnetic hull materials and onboard electronic systems, which have limited the conventional use of shipborne fluxgate magnetometers on cruise missions [7,8,9]. Despite these limitations, fluxgate magnetometers have been widely applied in various unmanned platforms such as UUVs (Unmanned Underwater Vehicles) and UAVs (Unmanned Aerial Vehicles) [4,10,11,12,13], due to their significant advantages of requiring small space and enabling continuous, unattended observation when mounted on vessels.

In addition to fluxgate magnetometers, various magnetic field sensing technologies exist, such as Hall-effect, Giant Magneto-Resistance, and Giant Magneto-Impedance sensors [14,15,16]. The magnetic sensors for marine magnetic survey require to measure high-resolution magnetic fields due to the ship’s dynamic motion changes and also require long-term stability and sensing of low-frequency magnetic fields below 0.01 Hz with a sensitivity of 1 nT or higher. Therefore, fluxgate sensors satisfy all these conditions are preferred [17,18,19].

USVs (Unmanned Surface Vehicles) offer advantages over UAVs by enabling longer-duration marine surveys and acquiring higher-resolution data due to their ability to operate on the surface of the sea. Furthermore, they mitigate the risk of losing expensive equipment like UUVs, can effectively explore shallow water areas, significantly reduce operating costs compared to existing manned research vessels, and can autonomously collect data for extended periods even in harsh marine environments, thus being regarded as an innovative platform that can greatly enhance the efficiency of marine surveys [20].

Therefore, USVs are increasingly being explored for high-precision marine magnetic surveys. While one common strategy to mitigate the inherent magnetic noise of a platform is to tow the magnetometer behind the USV to physically separate the noise [21], another approach is to mount the sensor on the vessel and to make the platform using paramagnetic materials to mitigate the magnetic interference. Mounting the sensor directly on the USV offers significant operational advantages compared to towed systems. Specifically, it eliminates the risk of the tow cable snagging on fishing boats and gear, aquaculture facilities, or seabed obstacles, thereby enhancing operational safety and reliability.

However, the integration of fluxgate sensors on USVs introduces unique technical challenges, particularly from electromagnetic noise generated by the vehicle’s electrical systems and navigation equipment. While paramagnetic materials, such as stainless-steel alloys, have been utilized in deep-tow magnetometer systems to mitigate platform-induced interference [22,23], their application to USVs remains relatively unexplored. Unlike traditional towed systems, USVs face distinct challenges that require innovative solutions for noise reduction.

Consequently, even with hulls constructed from paramagnetic materials, the array of electronic devices integrated into USVs, such as propulsion systems (motors, propellers), power systems (batteries), and navigation and communication equipment, can still introduce electromagnetic interference (EMI) that perturbs the surrounding magnetic field. Therefore, the precise measurement and compensation of magnetic field effects originating from the USV platform itself becomes essential for acquiring high-quality vector magnetic data.

To address this platform-induced magnetic interference and accurately correct the ambient geomagnetic field, various correction methods have been proposed on various platforms, including aircraft, ships, and unmanned systems [4,10,24,25,26]. The correction methodologies developed within these studies primarily focus on removing the permanent and induced magnetic effects associated with the platform’s attitude variations from the magnetic field measurements. However, each correction method utilizes a slightly different model and approach.

This study aims to evaluate the magnetic effect of a paramagnetic Unmanned Surface Vehicle (USV) itself and assess the utility of marine geomagnetic surveys using magnetic data acquired by a fluxgate magnetometer mounted on the vessel. To achieve this, we compare and analyze various magnetic compensation methods previously applied to different platforms, seeking the optimal methodology to reliably remove induced magnetic interference and operational noise generated by the paramagnetic USV. Ultimately, this research intends to contribute to expanding the application scope of high-precision marine vector magnetic surveys utilizing paramagnetic USVs.

## 2. Methods

In this study, magnetic data were collected approximately 1 km offshore from Chilpo Harbor in Pohang, southeastern Korean Peninsula (Figure 1). We mounted the MagDrone R3 fluxgate magnetometer system (Sensys, Bad Saarow, Germany) on the ship, consisting of two fluxgate magnetometers originally used in drones, and capable of acquiring a wide range of magnetic field (−75,000 nT to 75,000 nT) with a high sampling frequency (up to 250 Hz) and high resolution of 0.15 nT. We target geomagnetic field data (typically ranging from 30,000 to 60,000 nT) and magnetic anomaly (−2000 to 2000 nT). The measurement platform was the PARANG-5500 Unmanned Surface Vehicle (USV) manufactured by Ocean Tech (Goyang, Republic of Korea) (Figure 2). This USV features an autonomous navigation system capable of obstacle detection and employs multiple communication methods, including 2.4 GHz Wi-Fi, 900 MHz ISM band radio, and Iridium satellite communication.

The vessel’s hull is primarily aluminum alloy, offering several advantages for magnetic surveys. First, as a paramagnetic material, aluminum exhibits weak magnetic susceptibility, significantly reducing hull-induced interference in magnetic measurements. Second, aluminum provides excellent corrosion resistance and an optimal strength-to-weight ratio for marine operations. The stable magnetic properties of aluminum ensure long-term consistency in magnetic measurements, as the hull’s magnetization can be mitigated and remains stable throughout operations, unlike ferromagnetic materials that show both strong induced and permanent magnetization (see Appendix A). However, essential ferromagnetic components such as engines, generators and other equipment still contribute to the vessel’s magnetic signals. These magnetic influences cannot be ignored when correcting the USV-mounted magnetic data.

To correct for the USV’s magnetization effect, we collected five circular maneuver datasets with a 10 m radius, considering the vessel size and shallow depth (20–30 m) of the geomagnetic source. We also acquired data from the same track-line eight times to confirm the consistency of the geomagnetic anomalies after the correction (Figure 1b,c).

In this experiment, sensor 1 is positioned at the center of the mast to mitigate electromagnetic perturbations, using the mutual cancellation of magnetic fields at the bow and stern [3] and sensor 2 is positioned closer to the vessel’s electric motor (see Figure 2). This setup allowed us to compare the influence of magnetic field interference by the electric motor of the USV. The data is acquired with a 250 Hz sampling rate, while vessel attitude and position were recorded using onboard IMU sensors and DGPS at 50 Hz and 10 Hz, respectively.

## 3. Characteristics of Magnetic Signal

Before characterizing the magnetic signal acquired from the USV, we first analyzed ground magnetic data for comparison, collected on a beach with low human activity noise near the USV survey area (Appendix A). To understand inherent noise from the magnetometer system, we removed all electronic devices and magnetic materials from the operator’s body.

Prior to analyzing the characteristics of magnetic signals acquired from the USV, we first collected ground magnetic data from a low-noise beach near the USV survey area for comparison purposes (Appendix A). To understand inherent noise from the magnetometer system, the operator removed all electronic devices and magnetic materials during data collection.

Continuous Wavelet Transform (CWT) analysis results from both sensors were compared to identify their performance characteristics, confirming no significant difference in performance, as similar power levels were observed across both sensors. Additionally, strong power was detected in the low-frequency band of 0.001–0.003 Hz, which was identified as the primary geomagnetic signal targeted for analysis in this study. In the Fast Fourier Transform (FFT) and Short-Time Fourier Transform (STFT) analyses, strong signals were consistently observed at 60 Hz, which are considered noise generated by the magnetometer itself.

Next, we analyzed the magnetic signals recorded by Sensor 1 and Sensor 2 installed on the USV. Due to their different locations on the USV, the two sensors were affected by distinct noise sources, resulting in clearly distinguishable noise characteristics. To analyze signal characteristics, FFT analysis was performed to identify key frequency components by converting time-domain data to the frequency domain. The results showed that Sensor 2 exhibited generally higher power spectral density across the analyzed spectrum compared to Sensor 1. Furthermore, prominent EMI peaks were observed at approximately 30 Hz and 60 Hz in both sensors, along with common background noise, likely generated by electromagnetic components within the vessel and the magnetometer system (Figure 3a).

To examine the temporal variations of frequency components, STFT analysis was conducted. The results showed that Sensor 1 displayed relatively stable characteristics, featuring a persistent EMI band and weak transient signals. In contrast, Sensor 2, despite being the same sensor model, exhibited highly non-stationary noise with significantly elevated power levels concentrated in the 50–110 Hz band during periods when the vessel performed turning maneuvers (1300–3700 s) (Figure 3b). This strongly suggests the influence of operational noise originating from electrical systems near Sensor 2, particularly the electric motors.

For detailed time-frequency analysis, CWT analysis was employed. In the Sensor 1 data, strong signals were consistently observed primarily within the effective signal range inside the Cone of Influence (COI; indicated by the white dashed line in Figure 3c) in the frequency range below 0.003 Hz. Similar low-frequency characteristics were present in the Sensor 2 data. However, Sensor 2 additionally displayed periodic high-power features in the frequency range between approximately 0.005 Hz and 0.1 Hz. Consequently, the overall higher power levels observed in the Sensor 2 data indicate greater exposure to EMI from the vessel’s systems compared to the Sensor 1 location. Therefore, data from Sensor 1, having relatively less noise impact, were used for the subsequent magnetic correction analysis.

Marine magnetic surveys primarily measure the magnetic field resulting from the sum of the geomagnetic field originating from the outer core and the magnetic anomaly vectors within the crust. These magnetic field signals typically exist in very low frequency domains (<0.1 Hz) and do not exhibit rapid changes over short periods or distances under natural conditions. Therefore, based on the CWT analysis results, this study classified frequency components below 0.01 Hz as valid geomagnetic signals and those above this threshold as noise for subsequent analysis.

## 4. Data Processing

Data processing consists of three main procedures: preprocessing, ship’s attitude and magnetization correction, and post-processing. These procedures are detailed in the flow chart shown in Figure 4.

### 4.1. Preprocessing

As a preprocessing step before the ship’s magnetization correction, we applied a 4.5 Hz cutoff Butterworth low-pass filter to the original 250 Hz raw data before down-sampling it to 10 Hz. Considering the USV’s 1–3 knot operating speed, the resulting 0.12 m spatial resolution from 10 Hz sampling was sufficient for resolving geomagnetic anomaly signals in shallow waters of 20–30 m depth. This filtering process eliminates high-frequency noise and enhances the accuracy of subsequent correction procedures to reduce loss of signal quality.

To integrate data from asynchronous sensors, both IMU (50 Hz) and GPS (10 Hz) datasets were harmonized to a 10 Hz sampling rate through linear interpolation. Temporal synchronization between sensors was achieved by calculating the vessel’s azimuth from the magnetometer’s horizontal magnetic components and aligning it with GPS-derived headings and IMU yaw data through cross-correlation analysis.

### 4.2. Correction for the USV’s Attitude and Magnetization

The shipborne magnetic data are influenced by the vessel’s magnetic characteristics and its attitude changes [5,28]. These influences complicate data interpretation and contribute to the more prevalent use of towed scalar magnetometers, despite the advantages of fluxgate magnetometers in acquiring vector data [6,9,29]. Several methodologies have been proposed to correct for the magnetic influences on fluxgate magnetometers mounted on ferromagnetic vehicles [24,25,26]. In this study, we compare and analyze three mathematical approaches to determine which method is more accurate for correcting fluxgate magnetometer data on the USV. All proposed equations fundamentally originate from the same formulation, as represented in Equation (1) [8]. The first equation delineates the direct transformation relationship between sensor measurements ($H_{obs}$) and the ambient magnetic field ($H_{env}$)(1)$$H_{obs}=A\cdot R\cdot H_{env}+H_{p}+H_{v}$$
where A is a 3 × 3 transformation matrix reflecting the sum of Earth’s magnetic field and the ship’s induced magnetization susceptibility tensor, and R represents a rotation matrix for Roll, Pitch, and Yaw. $H_{env}$ is the Earth’s background magnetic field a three-component vector (3 × 1), $H_{p}$ is a 3 × 1 vector representing the ship’s permanent magnetization, and $H_{v}$ is the ship’s viscous magnetization (VM). The VM, commonly referred to as magnetic viscosity or magnetic aftereffect, occurs due to changes in magnetization direction caused by variations in vessel attitude and the Earth’s magnetic field on vessels, which represents the magnetic field generated by eddy currents within the hull. Although viscous magnetization is typically weak and does not significantly affect data interpretation [8], it can accumulate during long-term vessel navigation and be visible as long-wavelength signals; consequently, it reduces the reliability and precision of high-resolution magnetic survey results [3,9]. Therefore, to improve the accuracy of vector magnetic field measurements, it is very important to have a comprehensive understanding of the VM effect and develop hardware or software compensation strategies to reduce this effect.

Since the USV used in this study is constructed with aluminum alloy, a paramagnetic material that exhibits negligible intensities of VM, we approached the problem by eliminating this term from the equation.(2)$$H_{obs}=A\cdot R\cdot H_{env}+H_{p}$$

This equation directly expresses how the hull’s magnetic characteristics and attitude changes affect the measurements.

The equation proposed in [25] is a transformed version of Equation (2):(3)$$B\cdot H_{obs}+H_{pb}=R\cdot H_{env}$$
where B is A^−1^, and H_pb_ represents $-A^{-1}\cdot H_{p}$. Although these equations are mathematically equivalent, they differ in the process of optimizing the least squares parameters.

The formulation proposed by [26] is identical to that of [24] but includes an additional P matrix to correct for the non-orthogonality between the sensor and the platform. Furthermore, it presents a method for correction by calculating only the principal components (xx, yy, zz) of the susceptibility tensor A and the permanent magnetization component $H_{p}$.(4)$$H_{obs}=A\cdot P\cdot R\cdot H_{env}+H_{p}$$

In this study, we separated the USV’s 360-degree rotation intervals (Figure 1b) and analyzed data from each rotation section individually to correct the vessel’s magnetization using four different methods. The magnetometer records magnetic field data in a ship-fixed coordinate system, whereas the Earth’s magnetic field is expressed in the Earth coordinate system. When the USV rotates 360 degrees, the yaw angle varies from 0 to 360 degrees, while roll and pitch angles fluctuate within a relatively narrow range. Although this rotation pattern predominantly affects the x and y components of the surrounding geomagnetic field, strategic maneuvering of the USV can induce sufficient variations in the *z*-axis measurements. We can maximize the changes in the z-component measurements from the correction maneuver, enabling comprehensive correction of all three magnetic components through the least squares method. Since the magnetic data is not a square matrix, an inverse matrix does not exist. Therefore, we computed A and H_p_ from Equation (2) [24], and B and H_pb_ from Equation (3) [25] using the pseudo-inverse via Singular Value Decomposition (SVD) for the least squares method. For Equation (4) [26], we first calculated the non-orthogonality matrix, and then computed the principal components of the A matrix and the H_p_ array using the pseudo-inverse via Singular Value Decomposition (SVD) for the least squares method. The correction coefficients derived from each method are summarized in Table 1.

Each matrix A and B is a 3 × 3 matrix representing the sum of the susceptibility tensor of the vessel and Earth’s magnetic field vector components, where xx, yy, and zz components of the tensor should exhibit greater than 1 since the vessel has positive susceptibility. However, the two matrices A and B exhibit some discrepancies in each component.

To assess the corrected results from the two correction equations, we used the Normalized Root Mean Square Error (NRMSE) to quantitatively compare the corrected magnetic field data with the International Geomagnetic Reference Field (IGRF) model by the following equation.(5)$$NRMSE=\sqrt{\Sigma {\left(\frac{H_{corr}-H_{env}}{H_{env}}\right)}^{2}/N}$$
where $H_{corr}$ represents the corrected magnetic field vector component, $H_{env}$ denotes the corresponding component of the IGRF model, and N is the total number of data points.

NRMSE was compared separately for the horizontal and vertical components for each of the three correction equations: (2), (3), and (4) and only vessel attitude was corrected. The NRMSE values for the horizontal components show 0.0608, 0.0251, 0.037, and 0.0365, and for the vertical component, the values exhibit 0.0322, 0.0134, 0.0244, and 0.0224, respectively. This indicates that the correction using Equation (3) converges well to the ambient magnetic field (IGRF model) in both horizontal and vertical components. The corresponding correction results and their residual histograms are presented in Figure 5 and Figure 6, respectively.

These results can be explained by the structural advantages of Equation (3) when applying the least squares method. In Equation (3), the right-hand side consists entirely of the stable and well-defined background magnetic field values, typically derived from the IGRF or a stably obtained background magnetic field. This enables the independent optimization for the unknown variables (B, $H_{pb}$) on the left-hand side, enhancing numerical stability during the least squares fitting process. Comparing the method presented in [26] with that of [25], the equations used for correction are identical in all aspects except for the non-orthogonality correction matrix (P), which is excluded in [25]. However, the method in [26] performs the correction using only the principal components (xx, yy, zz) of the susceptibility tensor, whereas [25] employs all nine components of the tensor as variables. This suggests that utilizing not only the major but also the minor components of the susceptibility tensor may have the capability to eliminate not only the generally known magnetic characteristics of the platform but also other unknown linear features exhibiting low-frequency electromagnetic noise.

Therefore, we found that the correction approach based on Equation (3) achieved more accurate and reliable results compared to Equations (2) and (4). Consequently, we applied the correction coefficients obtained from Equation (3) to the entire dataset for comprehensive magnetic data correction (see Appendix A).

### 4.3. Post Processing

Following the correction for the USV’s attitude and magnetization, short-wavelength noise was still observed (Figure 7a) due to the unique technical characteristic of the fluxgate sensor exhibiting low sensitivities with short-wavelength wiggling noise. Also, VM from the ship’s attitude changes could be affected during data acquisition [9]. In ferromagnetic ships, VM are typically significant, resulting in magnetic field variations that are either very long-wavelength or appear constant during directional changes [3]. However, in the case of the Unmanned Surface Vehicle (USV), short-wavelength noise is dominantly observed during abrupt yaw variation (see Figure 3). This is likely due to the USV being constructed from a paramagnetic material, where instantaneous VM occurs during changes in vessel orientation, manifesting as short-wavelength noise.

These noise components can be removed through data filtering techniques. We first eliminated spiky noise exceeding ±100 nT from the main trend of the magnetic field. Subsequently, we applied a Butterworth low-pass filter with a 0.003 Hz cutoff frequency was applied to remove higher frequency noise artifacts. Then, by removing the vector components of the IGRF from the filtered magnetic field, we obtained the geomagnetic anomalies for each vector component.

## 5. Results and Discussion

This study demonstrated the advantages of employing an aluminum-alloy USV for shipborne magnetic surveys, including a significant reduction in magnetic interference compared to conventional ferromagnetic vessels. However, despite the use of a paramagnetic hull, the presence of residual ferromagnetic components, such as motors and other accessories, necessitates appropriate magnetic correction methods. The error histograms and NRMSE values clearly show that comprehensive correction accounting for both attitude and magnetization (i.e., using Equations (3) and (4)) yields significantly improved outcomes compared to attitude correction alone, even with the paramagnetic platform (Figure 5 and Figure 6).

In particular, the histograms of misfit against the IGRF model [30] illustrate the improved performance of Equation (3) [25] compared to the other methods (Figure 6). A key difference lies in their mathematical structure and optimization approach. Equation (2) [24] exhibited numerical instability, potentially due to its direct dependence on the noise containing observed magnetic field (H_obs_) and possible exclusion of non-orthogonality corrections. Equation (4) [26], structurally similar to Equation (2), integrates a non-orthogonality correction matrix (P) into Equation (2). However, this method only considers the principal components (xx, yy, zz) of the magnetic susceptibility tensor, neglecting the off-diagonal terms (xy, xz, yz). In contrast, Equation (3) [25] utilizes all nine components of the susceptibility tensor, providing a comprehensive model for induced magnetization described by the complete tensor. Furthermore, although Equation (3) does not explicitly combine the non-orthogonality correction matrix, its mathematical structure offers an advantage. From this perspective, the nine susceptibility tensor components are linear coefficients relating the induced field to the ambient field components. Consequently, solving for these coefficients inherently corrects not only for the nine components of the vessel’s susceptibility but may also automatically compensate for other noise sources exhibiting linear behavior with respect to the ambient field, potentially including non-orthogonality effects or linear drifts. This comprehensive linear correction likely contributes significantly to its improved performance compared to the other methods. Our comparative analysis demonstrates that Equation (3) [25], which utilizes all nine components of the susceptibility tensor and performs comprehensive linear correction, is the most robust method for simultaneously compensating for vessel attitude and induced magnetization. These findings suggest that even when using low-magnetic platforms, precise correction for both attitude and residual magnetization is essential to obtain accurate vector magnetic data.

To quantify the benefit of using this optimal correction method on the paramagnetic platform, we compared our USV results with reference data from the ferromagnetic R/V Kairei (KR05-17 dataset, JAMSTEC, Yokosuka, Japan). While direct field comparisons are complex due to differing times, locations, and vessel sizes, comparing platform-induced effects is informative as the fundamental hull susceptibility characteristics persist [7]. Applying Equation (3) revealed significantly more pronounced discrepancies between attitude-corrected and fully corrected anomalies for the ferromagnetic R/V Kairei (Appendix A). This is attributed to the strong induced magnetization of its hull, quantitatively supported by Kairei’s significantly higher susceptibility tensor compared to the paramagnetic USV, as estimated via Equation (3) (Table 1 and Appendix A). In contrast, these variations were relatively minor for the aluminium-alloy USV (Appendix A), although applying Equation (3) shows that it still significantly improved stability and consistency relative to attitude changes.

Focusing on the corrected USV data (Equation (3) applied), analysis along the entire track reveals significant variations due to residual high-frequency noise, even after the main correction (Figure 7a). This noise is particularly prominent during the vessel’s turning maneuvers and is thought to originate from vessel vibrations or attitude errors occurring during these turns. To compute the final geomagnetic anomalies suitable for interpretation, we subsequently removed spiky noise and applied an additional low-pass filter. This filtering resulted in final anomaly ranges of −310 to 340 nT, −1216 to 1827 nT, and −324 to 171 nT for the X, Y, and Z components, respectively (Figure 7a, red solid line).

Despite filtering, potential low-frequency discrepancies remain, particularly associated with maneuvers. To investigate consistency and residual effects, data from repeated tracklines (eight passes total) were analyzed (Figure 7b). Plotting the corrected data against latitude exhibits the generally consistent geomagnetic field throughout the transit path, confirming strong correction. However, track-line (e.g., latitudes 36.153–36.1535° N and 36.1565–36.157° N), different anomalies of the X and Y components were still observed. These residual effects are due to VM and are likely caused by eddy currents from the ambient magnetic field when the vessel turns rapidly. To prevent the generation of eddy currents, it is possible to use non-conductive materials for the hull, such as Glass Fiber Reinforced Polymer (GFRP) [21]. However, GFRP has no electromagnetic shielding effect, and it can perturb measuring magnetic fields highly susceptible to EMI from ferromagnetic parts or various equipment installed inside the hull, resulting in increased noise when measuring magnetic fields. Therefore, more complex shielding and grounding designs are required to resolve it. On the other hand, aluminum is a conductive material and it can effectively block EMI by the Faraday cage effect, resulting in simplifying the grounding design of research vessels that are equipped with a large number of precision measuring devices and increasing the stability of magnetic field measurements.

Meanwhile, the magnetometer’s inherent DC offset and drift are implicitly removed together during the correction process of the figure of rotation maneuvers. This is achieved by compensating for the vessel’s major magnetic effects (i.e., permanent and induced magnetization), which in turn converges the overall measured values to the background magnetic field. The resulting magnetic field cannot be fully corrected by the applied method (Equation (3)) and thus remains as a low-frequency signal. Notably, the VM effect on the paramagnetic USV disappears rapidly within about 50–100 m after a turn, corresponding to a relatively short temporal relaxation (approx. 0.5–1 min at 3 knots). This contrasts sharply with conventional ferromagnetic R/Vs (e.g., [31]), where eddy current effects persist much longer after direction changes (cf. Appendix A).

To understand the accuracy of the corrected and filtered fluxgate data along track segments unaffected by VM, we compared our results with high-resolution data from a crossing towed scalar magnetometer survey (Appendix A). The comparison shows good agreement in the overall trend and shape of the magnetic anomaly between the corrected fluxgate data and the towed scalar data (Appendix A), confirming that relative magnetic variations are reliably captured. However, a consistent offset of approximately 200 nT exists between the absolute anomaly values from the two systems, likely reflecting differences in absolute calibration or baseline levels between the distinct measurement systems and platforms.

The residual VM effect highlights that VM remains a significant challenge in shipborne fluxgate magnetic data processing even for paramagnetic platforms. The complexity arises from eddy currents and associated secondary fields generated within conductive (even paramagnetic) or residual ferromagnetic materials due to temporal variations in the primary magnetic field experienced by the vessel (e.g., attitude or azimuth changes) [8,32]. While methods exist to mitigate VM on ferromagnetic vessels using a towed scalar magnetometer [3,9], robust correction using only shipborne fluxgate data remains difficult.

Our results demonstrate that the paramagnetic hull significantly reduces the magnitude and duration of VM compared to ferromagnetic vessels (Appendix A), alleviating the problem without relying solely on signal processing. The observed short-wavelength noise during rapid yaw changes is consistent with weak, temporary paramagnetic magnetization. Although low-pass filtering removed localized high-frequency noise, software-based VM correction remains challenging, suggesting that a combined approach using appropriate platform materials and advanced correction techniques is optimal. This approach will be even more critical for future applications, as the recent development of higher-sensitivity sensors, such as those using Giant Magneto-Resistance or Magneto-Impedance, also makes them more susceptible to platform-induced magnetic noise. Therefore, the correction methodology validated in this study is considered essential to keep pace with these new sensor technologies. Future research applying and optimizing these algorithms for MI sensor data is expected to contribute significantly to the advancement of marine geophysical exploration technology.

## 6. Conclusions

This study demonstrated the feasibility of acquiring high-resolution shipborne fluxgate magnetic data utilizing a paramagnetic aluminum-alloy USV, which significantly reduces magnetic interference, particularly the duration of VM.

We confirmed that robust correction for vessel attitude and induced magnetization remains essential for accurate vector data, even with paramagnetic platforms. Specifically, a comparative analysis of existing correction methods (Equations (2)–(4)) revealed that Equation (3) [25] demonstrated the most effective correction performance.

Despite effective filtering of high-frequency noise, two key challenges requiring further attention were identified: residual low-frequency VM effects associated with eddy currents during vessel turns, and a consistent offset error (approximately 200 nT) when compared to towed scalar data. This offset presents an inherent limit for precise magnetic field measurements, even with advanced correction strategies. These persistent discrepancies highlight the challenges in obtaining reliable magnetic fields.

The methodologies validated here show potential for extending high-resolution vector magnetic surveys using paramagnetic unmanned platforms. Future work should focus on developing advanced VM correction techniques, optimizing sensor placement, and integrating complementary geophysical data. Additionally, research on the potential impact of the platform’s electrical conductivity characteristics on magnetic response should also be pursued. These efforts will ultimately contribute to significantly enhancing the accuracy and resolution of vector magnetic anomaly maps derived from unmanned platforms.

## Figures and Tables

**Figure 1 sensors-25-04511-f001:**
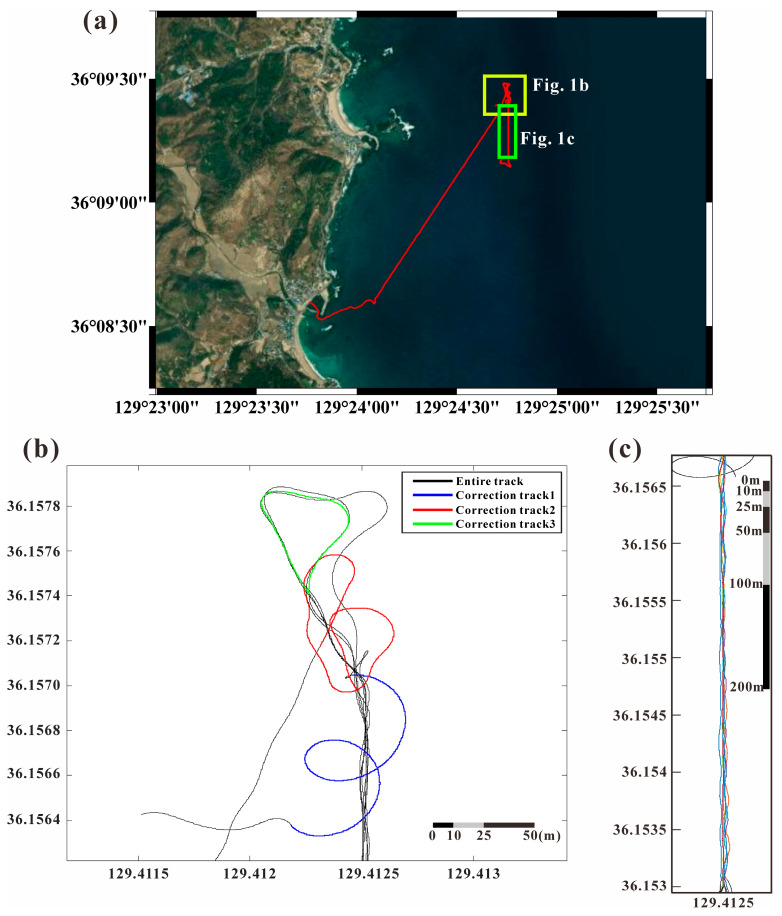
Survey area and USV (Unmanned Surface Vehicle) track line correcting for the ship’s magnetization. The USV conducted specific rotational patterns to correct the onboard fluxgate magnetometer and compensate for the vessel’s magnetizations. (**a**) A Satellite image [27] in the study area with the survey trackline of the USV. The yellow box indicates the area where the USV conducted the figure of rotation maneuvers for the ship’s magnetization correction (**b**). The green box depicts repeated tracklines in (**c**). (**b**) Detailed view of the figure of rotation maneuvers for the correction. The black solid line shows the complete track, and blue, red and green solid lines display each figure of rotation track (correction track 1, correction track 2, correction track 3). (**c**) Map view of the repeated survey tracklines run by the USV along the same path. Each color represents a different trackline.

**Figure 2 sensors-25-04511-f002:**
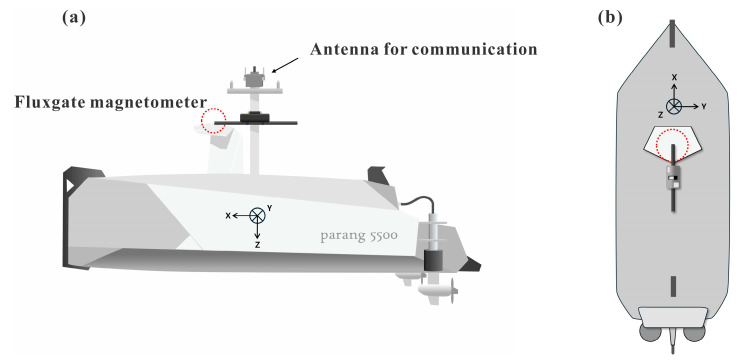
Schematic illustration of the USV (Parang 5500) showing the installation location of the fluxgate magnetometer (indicated by red dashed line). (**a**) Side view of the USV showing the main magnetic sensor (red dashed line). (**b**) Top view of the USV with the sensor position. The main magnetic sensor was mounted on top of the mast to mitigate magnetic interference from the vessel’s hull and electronic components.

**Figure 3 sensors-25-04511-f003:**
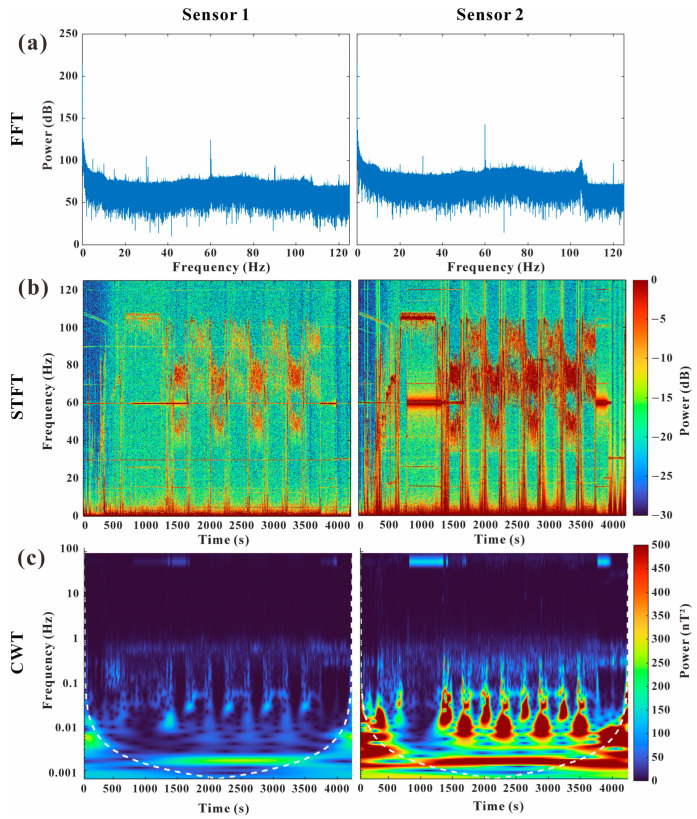
Frequency and time-frequency analysis (FFT, STFT, CWT) of magnetic data from Sensor 1 and Sensor 2, acquired and collected from the paramagnetic USV. (**a**) Fast Fourier Transform (FFT) spectrum in the frequency domain, showing the amplitude distribution across frequencies up to 125 Hz. The *y*-axis represents the amplitude on a logarithmic scale. (**b**) Short-Time Fourier Transform (STFT) spectrogram, illustrating the time-frequency distribution of the magnetic signal during the survey period. The color scale indicates signal intensity in decibels (dB), with red representing higher intensities and blue indicating lower intensities. (**c**) Continuous Wavelet Transform (CWT) in the time-frequency domain, displaying the time-frequency distribution of the magnetic signal during the survey period. The color scale represents signal intensity in (nT^2^) units, with red indicating stronger signals and blue representing weaker signals. The white dashed line in the CWT image represents the confidence interval of the analysis.

**Figure 4 sensors-25-04511-f004:**
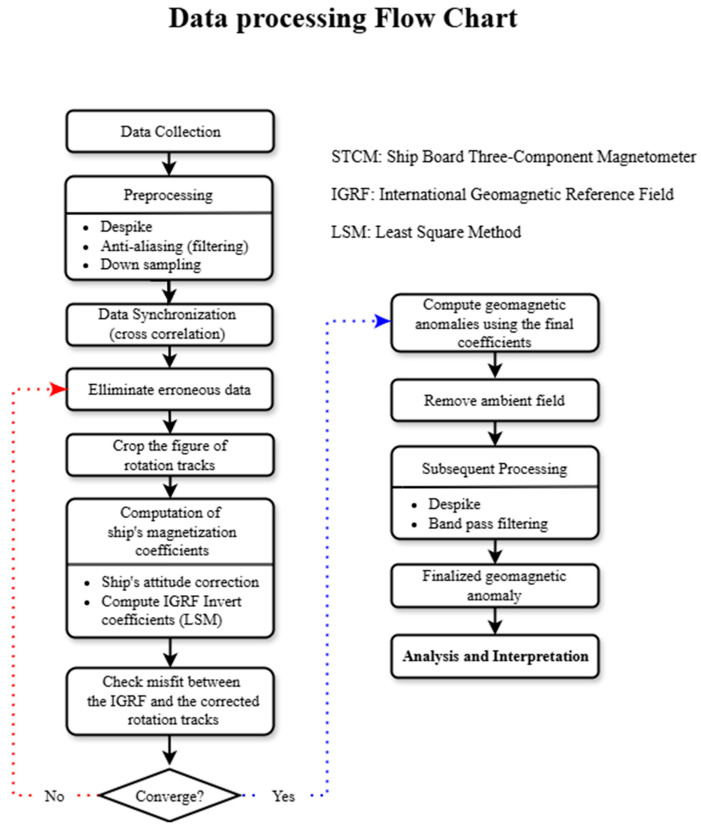
Magnetic data processing flow chart. The left branch shows the iterative process for computing correction coefficients using the data of rotation tracks, after the synchronization of the time delay. The convergence check ensures the reliability of the calibration coefficients. The right branch illustrates the subsequent processing steps after the matrix computation.

**Figure 5 sensors-25-04511-f005:**
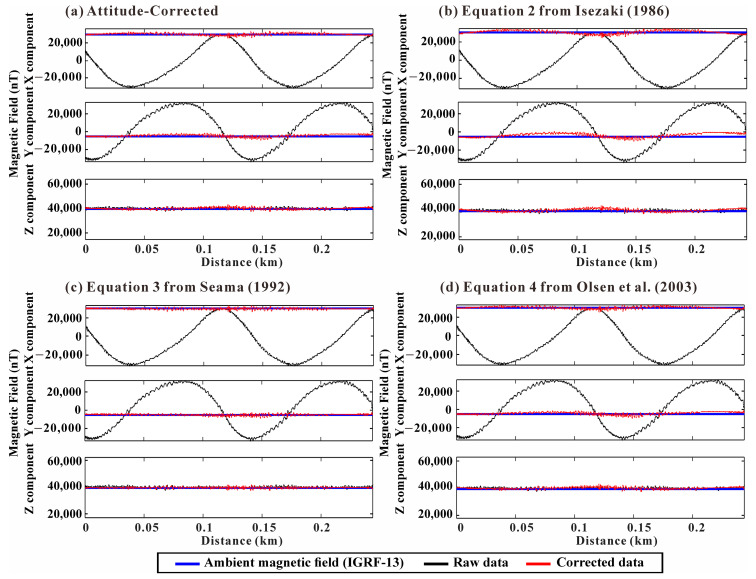
Comparison of correction results for correction maneuver track 1 (Figure 1b). (**a**) Ship’s attitude corrected. (**b**) Correction results from Equation (2) [24]. (**c**) Correction results from Equation (3) [25]. (**d**) Correction results from Equation (4) [26]. All equations are based on the same model formulation; they yield different results due to their different approaches to parameter optimization in the least squares method. Black: raw data; blue: ambient magnetic field (IGRF-13); red: corrected data.

**Figure 6 sensors-25-04511-f006:**
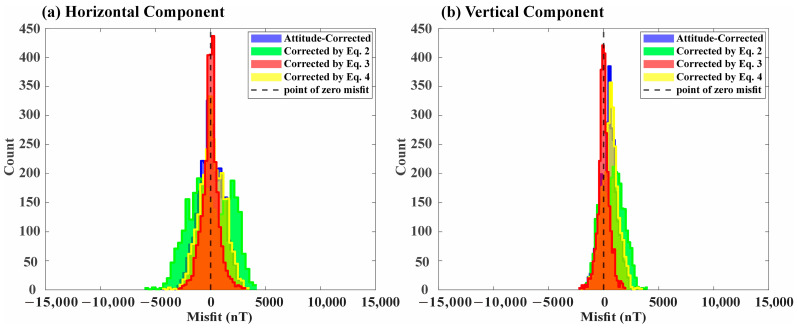
Histograms of the misfit between the ambient magnetic field (IGRF-13) and the corrected USV magnetic field for the horizontal and vertical components measured along rotation track 1. Four different approaches for correction were compared: correction using Equation (2) [24] (green), correction using Equation (3) [25] (red), correction using Equation (4) [26] (yellow) and correction based solely on vessel attitude variations (blue). The misfit histogram resulting from Equation (3) demonstrates the lowest misfit for both the horizontal components (combined x and y-components) and the vertical component (z-component).

**Figure 7 sensors-25-04511-f007:**
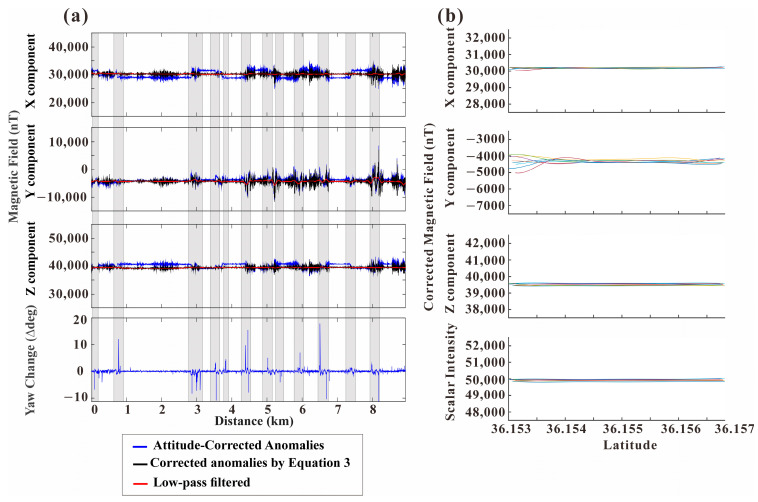
Magnetic field correction results for data acquired along repeated tracklines. (**a**) Results of correction steps along the entire survey track: attitude-corrected anomalies (blue), anomalies after applying magnetic correction (Equation (3); black), and the final anomalies after applying a low-pass filter (red solid line) for X, Y, Z components, along with Yaw change. Shaded areas indicate specific segments. (**b**) Corrected magnetic field components (X, Y, Z) and scalar intensity plotted against latitude for all repeated tracklines shown in Figure 1c, demonstrating the consistency achieved after applying the magnetic correction (Equation (3)). Each color corresponds to a specific trackline shown in Figure 1c.

**Table 1 sensors-25-04511-t001:** Twelve constants of the ship’s magnetization determined from the figure-of-rotation tracks using Equation (2) (up), Equation (3) (middle) and Equation (4) (down).

Correction coefficients from Equation (2)
$A=\left[\begin{array}{ccc} 0.9679 & -0.0188 & -0.6011\\ -0.0099 & 0.9549 & -0.4577\\ -0.0053 & 0.0059 & 0.1001 \end{array}\right]\;H_{p}=\left[\begin{array}{c} 22187.7\\ 17958.7\\ 35814.7 \end{array}\right]$
Correction coefficients from Equation (3)
$B=\left[\begin{array}{ccc} 1.0079 & 0.0465 & -0.3216\\ -0.0051 & 1.0342 & -0.1424\\ -0.0228 & 0.0013 & 1.0947 \end{array}\right]\;H_{pb}=\left[\begin{array}{c} 14143.3\\ 5436.44\\ -4308.16 \end{array}\right]$
Correction coefficients from Equation (4)
$A=\left[\begin{array}{ccc} 0.9959 & 0 & 0\\ 0 & 0.9968 & 0\\ 0 & 0 & 0.9993 \end{array}\right]\;P=\left[\begin{array}{ccc} 1 & 0 & 0\\ -0.0012 & 1 & 0\\ 0.0007 & 0.0015 & 1 \end{array}\right]$ $H_{p}=\left[\begin{array}{c} -19.36\\ 116.04\\ -128.84 \end{array}\right]$

## Data Availability

For data requests, please contact the corresponding author.

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
