# Peer review of "Magnetic Data Correction for Fluxgate Magnetometers on a Paramagnetic Unmanned Surface Vehicle: A Comparative Analysis in Marine Surveys"

_sensors, 2025, doi:10.3390/s25144511_

Round 1
Reviewer 1 Report
Comments and Suggestions for Authors
The authors well presented the comparative study results for data correction of USV-board fluxgate magnetometer. I have only a few comments and suggestions.
-It would be helpful for readers to explain the typical detection level required to identify magnetic anomalies (whether it is in the nT range or hundreds of nT range), and based on that, the performance required for the fluxgate magnetometer should be discussed. It is said in section 2 that the chosen fluxgate magnetometer has 0.1 nT precision, but those magnetometers tend to have DC offset drift against temperatures. It is helpful to readers if such DC offset drift can also be correct with the equation3 or remains as ambiguity like the effect from viscus magnetization.
-Regarding viscus magnetization, is it effective to change the material of vessel from aluminum to GFRPs so that the eddy current would be reduced? Is there any previous research discussing VM level found on non-conductive vessels?
-In section 3 , authors tested magnetic environment with two sensors (Sensor1 and Sensor2), but seems to be using data from only Sensor1. For future work, cross correlation between two sensors may be helpful to remove the effect from VM and reduce the residual absolute offset.
Author Response
Comment 1: It would be helpful to explain the detection level required to identify magnetic anomalies (e.g., in the nT or hundreds of nT range), and whether the fluxgate magnetometer's precision is sufficient. Also, address whether DC offset drift can be corrected using Equation (3) or remains ambiguous like viscous magnetization.
Response 1:
We thank the reviewer for their important comments.
The detection level for identifying a magnetic anomaly varies depending on the application. In geophysical surveys, the wavelength of magnetic anomalies from geological features range from tens of meters to hundreds of kilometers, and the typical measured geomagnetic field amplitude is between 30,000 to 60,000 nT. Since these anomalies are very large in scale but exhibit a low amplitude, a high-precision magnetic sensor is essential. We believe the fluxgate magnetometer (MagDrone R3) used in this study, which has a resolution of 0.15 nT and a measurement range of -100,000 to 100,000 nT, sufficiently meets these requirements. We have added this clarification to Lines 117-120 of the main text.
Due to the nature of fluxgate magnetometers, DC offset and drift can be corrected using data from a figure-of-eight correction, as described in Equation (3). This process removes the vessel's constant magnetic noise (DC offset) and linear trend magnetic noise (drift) by referencing the ambient geomagnetic field (IGRF). As the measurement data is made to converge to the ambient magnetic field, the DC offset and drift are naturally eliminated. These are errors that can be removed with appropriate correction techniques like Equation (3).
However, abrupt temperature changes during a survey can induce additional drift in the fluxgate sensor. Although fluxgate sensors are relatively stable against temperature variations, we consider the sensor to be stable in our case because the vessel's movement is not fast, and significant temperature changes do not occur rapidly during marine surveys. This point has been further described in Lines 467-471 of the manuscript.
Comment 2: Regarding viscous magnetization, would changing the vessel material from aluminum to GFRP help reduce eddy currents? Are there previous studies addressing VM levels in non-conductive vessels?
Response 2:
We agree with the reviewer's opinion. As pointed out, a GFRP (Glass Fiber Reinforced Polymer) vessel is an excellent alternative as its non-conductive material resolves the eddy current problem in principle. However, GFRP vessels can have unpredictable residual magnetic fields due to irregularly distributed ferromagnetic reinforcements within the hull.
On the other hand, while an aluminum hull does induce viscous magnetization (VM) through eddy currents, it has the significant advantage of effectively controlling internal electromagnetic interference (EMI) through its inherent Faraday cage effect. The VM induced by eddy currents is generally sensitive to changes in the vessel's azimuth. While the VM effect is very large in ferromagnetic vessels (Choe and Seama, 2024), it is relatively short-lived in an aluminum alloy. Therefore, conducting surveys while minimizing abrupt changes in azimuth can make aluminum a more advantageous choice. We added this in Lines 455-466.
Previous studies on viscous magnetization have primarily been reported in the fields of materials science, focusing on magnetic and metallic materials. Since the VM in the hull of a non-conductive vessel is very weak, we have not yet found studies that quantitatively analyze this characteristic.
Comment 3: Although two sensors were tested, only Sensor 1 data appears to be used. For future work, cross-correlation between the sensors could help reduce VM effects and residual offsets.
We appreciate the reviewer for the helpful comment. As mentioned in Figure 3 and the main text, we confirmed that the EMI from the ferromagnetic engine to sensor 2 with respect to sensor 1 is more significant through wavelet analysis. Therefore, we mainly used the data collected by sensor 1 during the main analysis. In addition, since the two fluxgate sensors located in different locations are affected by different ship magnetizations (i.e., VM, induced magnetization, permanent magnetization), it is difficult to reduce the influence of VM or residual offset through cross-correlation. If additional sensors are installed on the ship to characterize the ship's magnetization, the reviewer's suggestion may be applicable.
There are several attempts to remove the VM influences by simultaneously operating the towed magnetometer and the shipboard magnetometer and then compensating them, as shown below. It is already described in the text (L491-492) and added to the reference list.
Choe, H., Seama, N., 2024. A New Correction Method for Ship’s Viscous Magnetization Effect on Shipboard Three-component Magnetic Data Using a Total Field Magnetometer. Geophysics and Geophysical Exploration 27 (2), 119–128. https://doi.org/10.7582/GGE.2024.27.2.119.
Choe, H., Dyment, J., So, B.-D., 2021. A crossover error correction algorithm for sea-surface marine magnetic data: Application to Northwest Pacific. Journal of Geological Society of Korea 57 (1), 67–77. https://doi.org/10.14770/jgsk.2021.57.1.67.
Reviewer 2 Report
Comments and Suggestions for Authors
In this paper, the magnetic interference significantly were reduced by using paramagnetic hull, while the maneuver-related high-frequency noise were significantly suppressed using low-pass filtering. A correction approach utilizing all nine components of the susceptibility tensor largely improved stability and accuracy. This study underscores the viability of utilizing paramagnetic USVs for conducting comprehensive vector magnetic surveys, highlighting advantages in VM mitigation and emphasizing the need for further development in VM correction and understanding system calibration offsets for achieving highest precision. The articles are well-written and the results are of great help to researchers in related fields. The paper can be published as it is.
Author Response
Comment 1: “The articles are well-written and the results are of great help to researchers in related fields. The paper can be published as it is.”
Response 1:
We sincerely thank the reviewer for the positive evaluation and encouraging comments. We are glad to know that the manuscript is considered well-written and relevant to the field. We appreciate your recommendation for publication.
Reviewer 3 Report
Comments and Suggestions for Authors
Submitted manuscript is devoted to rather narrow area of applications of magnetic field sensors. However, the subject is important and it is in the score of the MDPI Sensors. Very clear disadvantage of the manuscript structure (and general approach employed in it) is the absence of the clear connection with comparative analysis related to other types of navigation magnetic field sensing devices and wide experience databases on fluxgate sensor applications. In fact, the references list does not mention main fluxgate contributions to the field (see examples: Nielsen, O.V.; Petersen, J.R.; Primdahl, F.; Brauer, P.; Hernando, B.; Fernández, A.; Merayo, J.M.G.; Ripka, P. Development, construction and analysis of the Orsted fluxgate magnetometer. Meas. Sci. Technol 1995, 6, 1099–1115; Magnes, W.; Oberst, M.; Valavanoglou, A.; Hauer, H.; Hagen, C.; Jernej, I.; Neubauer, H.; Baumjohann, W.; Pierce, D.; Means, J.; Falkner, P. Highly integrated front-end electronics for spaceborne fluxgate sensors. Meas. Sci. Technol 2008, 19, 115801–115814, etc.)
Nowadays there are different kinds of magnetic field detectors for vehicle control. They are employing various magnetic effect such as Hall effect, giant magneto-resistance, magneto-impedance and others. The last one was shown to extremely sensitive with respect to the applied field and suitable for the magnetic noise reduction (see examples: Nishibe, Y., Ohta, N., Tsukada, et al., Sensing of Passing Vehicles Using a Lane Marker on Road with a Built-in Thin Film MI Sensor and Power Source, IEEE Transac. Vehic. Techn., 2004, vol. 53, no. 6, 1827–1834; E. Fernández, A. García-Arribas, J. M. Barandiarán, A. V. Svalov, G. V. Kurlyandskaya, C. P. Dolabdjian Equivalent Magnetic Noise of Micro-Patterned Multilayer Thin Films Based GMI Microsensor, IEEE SENSORS JOURNAL, 15(11), (2015) 6707, etc.). It would be important to provide general comparison of the existing devices/prototypes parameters with the obtained here keeping in mind that some of the magnetic materials (amorphous ribbons dopped with Mo or Cr) are well adapted to the marine environment being the basis for different types of the sensitive elements (for instance, both the flux gate and magnetic impedance).
The abstract is too long for the regular contribution, it significantly higher the usual length of 200 words. It also contains not very typical usage of the particular terms without proper definitions. For instance, viscous magnetization effects, are usually known as magnetic viscosity or after-effect, i.e. the slow relaxation of magnetization in magnetic materials over time. In contrary, the Introduction is rather short and focused on simply mentioning of technical developments without sufficient physical background. It must be extended and supported by the data related to other types of magnetic field sensors suitable for vehicle applications.
Figures have technical deficiencies: all symbols must be readable, i.e. the font size should be close to the font size of the figure captions.
Conclusions contain parts which should be moved to the discussion section.
The main idea of the study is that the paramagnetic hull combined with the correction shows the benefit. However, special attention should be paid to the problem of the variation of the conductivity as the navigation systems may have different dynamic responses.
Manuscript has a number of misprints (lost intervals between the number and the unit, variations of the format of the references, bold parts of the text, etc.). There is no Contributions section required by the journal.
Author Response
Comment 1: Submitted manuscript is devoted to rather narrow area of applications of magnetic field sensors. However, the subject is important and it is in the score of the MDPI Sensors. Very clear disadvantage of the manuscript structure (and general approach employed in it) is the absence of the clear connection with comparative analysis related to other types of navigation magnetic field sensing devices and wide experience databases on fluxgate sensor applications. In fact, the references list does not mention main fluxgate contributions to the field (see examples: Nielsen, O.V.; Petersen, J.R.; Primdahl, F.; Brauer, P.; Hernando, B.; Fernández, A.; Merayo, J.M.G.; Ripka, P. Development, construction and analysis of the Orsted fluxgate magnetometer. Meas. Sci. Technol 1995, 6, 1099–1115; Magnes, W.; Oberst, M.; Valavanoglou, A.; Hauer, H.; Hagen, C.; Jernej, I.; Neubauer, H.; Baumjohann, W.; Pierce, D.; Means, J.; Falkner, P. Highly integrated front-end electronics for spaceborne fluxgate sensors. Meas. Sci. Technol 2008, 19, 115801–115814, etc.)
Response 1:
We thank the reviewer for recognizing the relevance and importance of our topic and for providing valuable feedback on the manuscript's structure.
We agree that our study focuses on a specialized topic. Our primary goal is to evaluate and compare correction methodologies for magnetic data acquired from a non-magnetic USV, rather than to conduct a comparative review of different sensor technologies.
However, we fully agree with the reviewer’s critical point that our manuscript lacked references to major foundational studies in the fluxgate sensor field. We are grateful for the specific examples provided. In the revised manuscript, we have now incorporated these key references (e.g., Nielsen et al., 1995; Magnes et al., 2008) into our Introduction. This has significantly improved the context of our work by situating it within the broader history of fluxgate magnetometer development. We have also added a brief mention of other sensor types to clarify the rationale for focusing on the fluxgate sensor in this study (Lines 58-65).
Comment 2: Nowadays there are different kinds of magnetic field detectors for vehicle control. They are employing various magnetic effect such as Hall effect, giant magneto-resistance, magneto-impedance and others. The last one was shown to extremely sensitive with respect to the applied field and suitable for the magnetic noise reduction (see examples: Nishibe, Y., Ohta, N., Tsukada, et al., Sensing of Passing Vehicles Using a Lane Marker on Road with a Built-in Thin Film MI Sensor and Power Source, IEEE Transac. Vehic. Techn., 2004, vol. 53, no. 6, 1827–1834; E. Fernández, A. García-Arribas, J. M. Barandiarán, A. V. Svalov, G. V. Kurlyandskaya, C. P. Dolabdjian Equivalent Magnetic Noise of Micro-Patterned Multilayer Thin Films Based GMI Microsensor, IEEE SENSORS JOURNAL, 15(11), (2015) 6707, etc.). It would be important to provide general comparison of the existing devices/prototypes parameters with the obtained here keeping in mind that some of the magnetic materials (amorphous ribbons dopped with Mo or Cr) are well adapted to the marine environment being the basis for different types of the sensitive elements (for instance, both the flux gate and magnetic impedance).
Response 2:
We thank the reviewer for their very insightful suggestions that help place our work in the context of emerging technologies.
As you pointed out, we agree that next-generation sensors, particularly MI sensors, offer very high sensitivity. Your point that specific materials like amorphous ribbons are suitable for marine applications in both MI and fluxgate sensors is an excellent and important observation.
While our study's scope is focused on evaluating correction techniques for fluxgate sensors as the current industry standard, we believe your comment highlights a crucial direction for future research. Inspired by your suggestion, we have added a new paragraph to the Discussion section. This new text explicitly discusses how the comprehensive correction methods validated in our study will be even more critical for the future deployment of high-sensitivity sensors like MI, whose performance can be more susceptible to platform-induced noise.
We believe this addition significantly strengthens the paper's forward-looking implications. We thank you again for the valuable suggestion, which has added depth to our study.
Comment 3: The abstract is too long for the regular contribution, it significantly higher the usual length of 200 words. It also contains not very typical usage of the particular terms without proper definitions. For instance, viscous magnetization effects, are usually known as magnetic viscosity or after-effect, i.e. the slow relaxation of magnetization in magnetic materials over time. In contrary, the Introduction is rather short and focused on simply mentioning of technical developments without sufficient physical background. It must be extended and supported by the data related to other types of magnetic field sensors suitable for vehicle applications.
Response 3:
We appreciate the reviewer's valuable feedback.
Following the journal's guidelines, we have condensed the abstract to be under 200 words. Additionally, for technical terms such as viscous magnetization, we have added supplementary explanations to aid the reader's understanding. This content has been added to Lines 266-267 and Lines 272-275 of the main text.
Comment 4: Figures have technical deficiencies: all symbols must be readable, i.e. the font size should be close to the font size of the figure captions.
Response 4:
We thank the reviewer for pointing this out.
In response to the comment regarding the legibility of symbols and labels in the figures, we have adjusted the font size in all figures to ensure clarity.
Comment 5: Conclusions contain parts which should be moved to the discussion section.
Response 5:
We thank the reviewer for this structural suggestion. We have moved the relevant interpretative and comparative analysis from the Conclusions section to the Discussion, ensuring a clearer separation between results interpretation and final summary statements in Lines 411-416 of the main text.
Comment 6: The main idea of the study is that the paramagnetic hull combined with the correction shows the benefit. However, special attention should be paid to the problem of the variation of the conductivity as the navigation systems may have different dynamic responses.
Response 6:
Thank you for your valuable feedback.
We agree that the platform's electrical conductivity can affect eddy current induction and magnetic field when the magnetic field changes, which could lead to differences in the dynamic response of the navigation system.
However, the main focus of this study is to show that even when using a paramagnetic-based USV platform, magnetic components from ferromagnetic materials (like the steel engine) and time-dependent magnetization phenomena like viscous magnetization (VM) can still occur. This suggests that additional correction techniques are necessary for precise magnetic measurements.
Nevertheless, your point about the potential impact of conductivity differences between platforms on the magnetic response is an important consideration for future research. Accordingly, we have mentioned the need for follow-up studies to experimentally examine these differences in platform characteristics in Lines 516-518 of the main text.
Comment 7: Manuscript has a number of misprints (lost intervals between the number and the unit, variations of the format of the references, bold parts of the text, etc.). There is no Contributions section required by the journal.
Response 7:
We sincerely thank the reviewer for pointing out the formatting issues and a missing section. We have carefully proofread the manuscript and addressed all formatting inconsistencies, including proper spacing between numbers and units, consistent reference formatting, and the removal of unnecessary bold text. Additionally, we re-checked the entire manuscript to ensure full compliance with the MDPI formatting guidelines. Also we added “Author Contributions" section to the end of the manuscript following MDPI's author guidelines.
Round 2
Reviewer 3 Report
Comments and Suggestions for Authors
Work was improved up to the necessary level and it can be published in the present state.